# OpenReview forum: "Can Safety Fine-Tuning Be More Principled? Lessons Learned from Cybersecurity"
_NeurIPS.cc/2024/Workshop/SafeGenAi — SafeGenAi Poster_

### Official Review · Reviewer_QsEm · 2024-10-08

**Rating:** 5
**Confidence:** 3

**Review:**

This paper provides a comparison between LLM attacks and defenses to cybersecurity, arguing that knowledge from cybersecurity can be used to build stronger LLM defenses. The paper includes several analogies between LLM safety terms and their cybersecurity counterparts, as well as several examples.

Strengths:
 - The topic is of interest to the community and insights from a different field such as cybersecurity are appreciated.
 - The paper is well written and presented.

Weaknesses / Questions:

 - One of the main concerns is while the paper provides several analogies about how LLM safety can be put into a cybersecurity context, it does not talk in detail about how cybersecurity defenses can be applied as a more general defense against a wide range of attacks, if possible. I would have liked to see more discussion about how such insights from cybersecurity defenses can be applied here.
 - Since cybersecurity defenses can be applied at every level of a program, can the authors provide insights on how defenses can be "stacked" within an LLM (applied in the pretraining phase, fine-tuning stage, and/or at inference time)?
 - Are there cybersecurity defenses that can provide probabilistic guarantees for LLM safety? If so, how can they be applied?